# FIAEPI-KD: A novel knowledge distillation approach for precise detection of missing insulators in transmission lines

**Hanzhi Cui**[1], **Dawei Huang**[ID][2], **Wancheng Feng**[2], **Zhengao Li**[2], **Qiuxue Ouyang**[1], **Conghan Zhong**[ID][1]*

**1** College of Computer Engineering, Qingdao City University, Qingdao, China, **2** School of Intelligent Equipment, Shandong University of Science and Technology, Tai'an, China

* conghan.zhong@qdc.edu.cn

**Data availability statement:** All relevant data include two parts. Data is available from figshare, doi is 10.6084/m9.figshare.28925159.

## Abstract

Ensuring transmission line safety is crucial. Detecting insulator defects is a key task. UAV-based insulator detection faces challenges: complex backgrounds, scale variations, and high computational costs. To address these, we propose FIAEPI-KD, a knowledge distillation framework integrating Feature Indicator Attention (FIA) and Edge Preservation Index (EPI). The method employs ResNet and FPN for multi-scale feature extraction. The FIA module dynamically focuses on multi-scale insulator edges via dual-path attention mechanisms, suppressing background interference. The EPI module quantifies edge differences between teacher and student models through gradient-aware distillation. The training objective combines Euclidean distance, KL divergence, and FIA-EPI losses to align feature-space similarities and edge details, enabling multi-level knowledge distillation. Experiments demonstrate significant improvements on our custom dataset containing farmland and waterbody scenarios. The RetinaNet-ResNet18 student model achieves a 10.5% *mAP* improvement, rising from 42.7% to 53.2%. Meanwhile, the Faster R-CNN-ResNet18 model achieves a 7.4% *mAP* improvement, rising from 42.7% to 50.1%. Additionally, the RepPoints-ResNet18 model achieves a 7.7% *mAP* improvement, rising from 49.6% to 57.3%. These results validate the effectiveness of FIAEPI-KD in enhancing detection accuracy across diverse detector architectures and backbone networks. On the MSCOCO dataset, FIAEPI-KD outperformed mainstream distillation methods like FKD and PKD. Ablation studies confirmed FIA's role in feature focus and EPI's edge difference quantification. Using FIA alone increased RetinaNet-ResNet50's *mAP* by 0.9%. Combined FIA+EPI achieved a total 3.0% improvement, the method utilizes a lightweight student model for efficient deployment, providing an effective solution for detecting insulation defects in transmission lines.

## Introduction

In modern power transmission systems, transmission lines are crucial for power delivery. Their stable operation is essential for the smooth functioning of social production and daily

As for the COCO dataset, it can be obtained from its official website at https://cocodataset.org/#download.

**Funding:** The author(s) received no specific funding for this work.

**Competing interests:** The authors have declared that no competing interests exist.

life. Power accident statistics show that transmission line faults, like insulator absence, caused over 200 large-scale power outages in the past five years. These outages resulted in direct economic losses of hundreds of millions of yuan. Insulators are core components of transmission lines. They support and insulate the lines, which is vital for their electrical and mechanical stability. If insulator absence occurs, the risk of short-circuits, leakage, and other accidents increases significantly. This leads to large-scale power outages, which affect industrial production and residents' lives. It may also cause serious safety incidents, resulting in huge economic losses and social impacts. Therefore, detecting insulator absence accurately and efficiently has become an urgent need for the power industry to ensure transmission safety [1].

In recent years, inspection technology using Unmanned Aerial Vehicles (UAVs) has been widely used to detect insulators on transmission lines. UAVs are flexible and efficient. They can quickly capture image data of transmission lines, improving detection efficiency. However, as shown in Fig 1, the images captured by UAVs have several problems. The backgrounds are complex, with interference from trees and buildings [2]. This affects the performance of object-detection models and reduces accuracy. The scale of insulators also varies greatly in the images due to differences in shooting angles and distances. This makes detection more difficult. Additionally, deep-learning-based object-detection models have advantages in accuracy. However, they require large amounts of computing resources and storage space. They also have high hardware requirements, making them unsuitable for real-time deployment in resource-constrained environments.

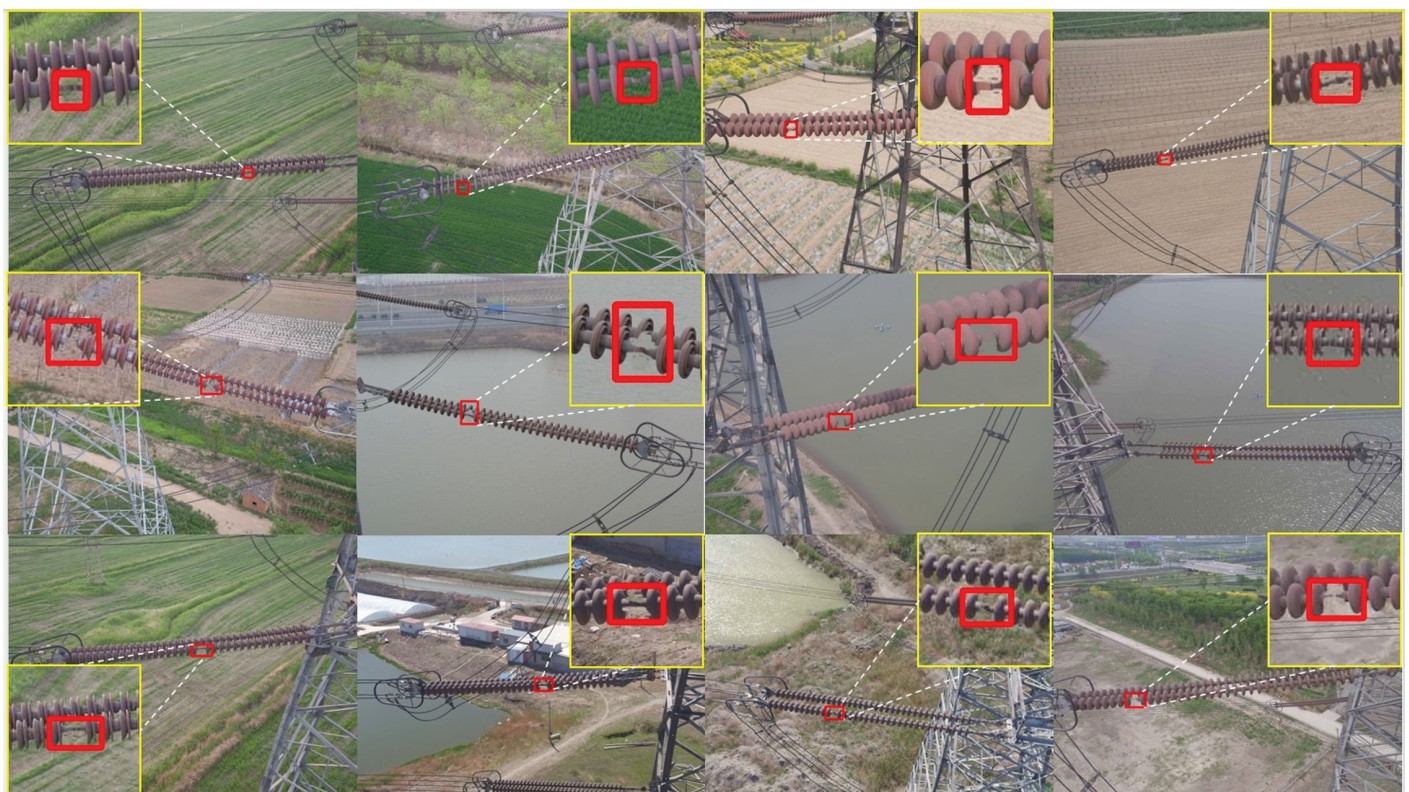

**Fig 1. The missing situation of transmission line insulators from multiple perspectives, with complex backgrounds covering different geographical environments such as farmland, water bodies, and factories.** The red box in the figure indicates the location of the missing insulator, which is used to highlight the target to be detected.

Currently, methods for detecting insulators on transmission lines can be divided into traditional image-processing-based methods and deep-learning-based methods [3]. Traditional image-processing methods, such as those based on color intensity profile transformation and machine learning [4,5], collect images through UAVs to build datasets [6]. These methods identify feature points, perform transformation processing, and use algorithms like decision trees and random forests for classification [7,8]. These methods work well in simple scenarios. However, they are affected by background, angle, and distance. In complex backgrounds, they may misjudge other objects as insulators. They also struggle to detect small or occluded insulator defects. On the other hand, deep-learning-based methods, such as the improved Cascade R-CNN algorithm, have improved detection accuracy. This algorithm combines pre-training strategies, FPN modules, and specific networks for detection. However, when handling complex-background images from UAVs, these methods still face challenges. They have low detection accuracy for small-target insulators and poor adaptability to insulators of different scales. Additionally, the high computational complexity of deep-learning models and their strict hardware requirements limit their application in resource-constrained scenarios.

Knowledge distillation technology [9] has been widely studied in object detection and can improve model performance [10]. However, when applied to insulator absence detection, existing knowledge distillation methods face many challenges. Traditional methods transfer features directly or use fixed semantic features. These methods cannot fully explore the unique features of insulators in complex backgrounds. As a result, the student model has difficulty learning key knowledge from the teacher model. This affects detection accuracy and the robustness of the model.

In summary, existing methods have various problems in insulator absence detection. These issues seriously restrict the safety monitoring of transmission lines. To address these issues, this paper proposes an innovative knowledge distillation strategy, FIAEPI-KD. This method combines attention guidance and edge detection. It uses the ResNet backbone network combined with FPN for multi-scale feature extraction. This captures feature information of insulators at different resolutions. The FIA module is designed to focus on the edge details of insulators. It enhances the perception of subtle features. The EPI module is introduced to quantify edge feature differences between the student and teacher models. It effectively guides the model to learn accurate edge representations. During the model training stage, an objective function is constructed. This function includes Euclidean distance, KL divergence, and losses based on FIA and EPI. It helps the model focus on feature space similarity and edge information retention, improving the accuracy and reliability of insulator absence detection.

The main contributions of this paper are as follows:

1. We proposed a novel knowledge distillation method for feature indicator edge-preserving detection. We applied it to insulator absence detection. This method provides a new way to improve the detection accuracy of insulator absence under different backgrounds.

2. We proposed the Feature Indicator Attention mechanism. This mechanism helps the lightweight model focus on key feature regions of insulator image data. It enables the model to accurately extract effective features.

3. We proposed the Edge Information Preservation module. This module enhances the detection model's attention to the edge information of insulators. It improves the accuracy of insulator absence detection.

4. We proposed a scalable framework for knowledge distillation. This framework integrates multi-scale feature fusion and edge-awareness. It enhances the performance of student models in resource-constrained environments. It also enables real-time detection of insulator defects across various transmission line conditions, improving both detection accuracy and efficiency.

## Related work

### Detection in power transmission scenarios

Existing methods for detecting insulators on transmission lines mainly fall into two categories: traditional image-processing-based methods and deep-learning-based methods.

For traditional methods, Han et al. [11] created the "InST detection" dataset and proposed a ResNet50-based multi-branch convolutional network. They addressed background interference, angle/distance impacts, and multi-fault detection using k-means priors and custom loss functions. Tomaszewski et al. [12] introduced a color intensity profile method with UAV-collected images. Features were transformed via Welch/Periodogram algorithms and classified using decision trees, random forests, or XGBoost.

In the field of deep-learningbased methods, there are many new ideas. Zhang et al. [13] enhanced Cascade R-CNN by integrating MSCOCO pretraining, FPN, and ResNeXt-101, with data augmentation (rotation/filtering). Siddiqui et al. [14] designed a CNN system for 17 insulator types, combining rotation normalization, ellipse detection, and gunshot defect analysis. Rahman et al. [15] developed a UAV-based system using LapSRN (super-resolution), LIME (low-light enhancement), BRISQUE (quality assessment), and fine-tuned YOLOv4. Zhai et al. [16] proposed MGRN (based on Faster R-CNN) with Appearance/Parallel/Spatial Geometric Reasoning modules for geometric defect detection. Liu et al. [17] devised a cross-modal enhancement algorithm (HICT for GAN-based image generation, MMA for multi-scale fusion) for UAV defect detection. Wang et al. [10] optimized YOLOv8 with deformable convolutions (C2f DCNv3), re-parameterized RCSP fusion, and a dynamic detection head for multi-defect recognition. Xia et al. [18] replaced ResNet-50 with MobileNet V1 in CenterNet, added CBAM attention and transposed convolutions, and validated on ID/CPLID datasets.

Lin et al. [19] proposed an Enhanced Disentanglement Module (EDM). This module extends feature disentanglement from the detection head to the feature pyramid network. It separates feature maps for classification and regression tasks to reduce task conflicts. Experiments show EDM improves accuracy by 2% on general datasets like COCO. However, its disentanglement relies on static feature separation. It does not dynamically adapt to complex backgrounds (e.g., farmland and water interference in transmission line images). The response alignment strategy (RAS) requires post-processing to filter predictions. It fails to explicitly suppress noise during feature extraction. Zhang et al. [20] adjusted the ROIAlign stride and redesigned the mask head. They fused global semantics with local details to improve large-object segmentation accuracy. However, the method uses fixed stride and pyramid-level fusion. This static strategy cannot adapt to dynamic scale changes (e.g., near-far insulator size differences) or edge blur in transmission line scenarios. The fusion mechanism ignores gradient sensitivity of edges. This may cause small-target details to be lost.

Ragini et al. [21] proposed SKDRMNet, a selective dense residual M-shaped network. It jointly removes rain streaks and haze to enhance object detection accuracy. SKDRMNet focuses on denoising and image quality restoration. However, it struggles to preserve edge details and precise localization in complex backgrounds (e.g., insulator defects in transmission lines). Chintakindi et al. [22] designed a lightweight network using depthwise separable convolution. It reduces parameters and suppresses background noise via feature enhancement. However, the method targets only pedestrian detection. It does not address feature conflicts between classification and regression tasks (e.g., insulator missing detection in noisy transmission line backgrounds). Chintakindi et al. [23] proposed SSAD, a single-shot multi-scale detection network. SSAD uses spatial feature relationships and global attention to highlight target regions. However, its fixed hierarchical feature fusion cannot adapt to abrupt insulator scale changes. Static attention mechanisms lose small-target features.

High-frequency noise (e.g., farmland textures and metal towers) disrupts SSAD's global spatial modeling. This leads to false detection of background artifacts as insulator edges. Gumma et al. [24] used a Binary Domain Selection Mechanism (BDSM) to enhance key regions. Nuthi et al. extracted frequency-domain features via wavelet transforms. These methods cause frequency confusion in complex backgrounds (e.g., dense wires and vegetation in transmission line images). For example, high-frequency farmland textures may be misclassified as insulator edges. Low-frequency interference (e.g., water reflections) cannot be removed by fixed filters.

Zhao et al. [25] proposed CWT-CARS-CNN. It enhances spectral sensitivity via continuous wavelet transform (CWT) and selects key bands via CARS. The method achieves high accuracy in coal-derived carbon estimation. However, CARS uses fixed statistical criteria for band selection. It ignores dynamic frequency noise (e.g., farmland and water in transmission line images). Key edge features (e.g., cracks) may be filtered out. The method lacks spatial attention to handle insulator scale variations caused by shooting angles. Zhang et al. [26] proposed a hybrid optimizer (WSHO). It combines Wingsuit Flying Search and Spotted Hyena Optimizer to select features (e.g., NDVI and SRre). A triple classifier (SVM-RF-BiLSTM) is used for mineral exploration. However, WSHO relies on artificial vegetation indices. Insulators and metal towers have similar textures in transmission line images. This causes feature confusion and false alarms. The Bi-LSTM's sequential modeling responds slowly to rapid scale changes of small insulators. It cannot balance speed and accuracy.

Most existing works primarily focus on general insulator or fault detection, overlooking the specific challenge of detecting missing insulators. This task is particularly difficult as it often involves small, irregular, and context-sensitive regions. The complexity of UAV images, including factors like distant views, varying scales, weather conditions, and structural occlusions, significantly complicates the accurate localization of missing insulators.

Although various object detection models such as YOLO, RetinaNet, and Faster R-CNN have been applied to insulator detection, they often struggle to detect the subtle missing parts of insulators, especially in conditions with haze or weak illumination. Moreover, while recent advancements aim to support real-time deployment with lightweight detectors, these models often trade off spatial sensitivity and multi-scale representation to reduce computational costs. Thus, insulator defect detection remains a challenging task due to the limitations of current detection methods in handling complex backgrounds and fine details.

## Knowledge distillation

There are also various types of knowledge distillation methods. Ren et al. [27] proposed a Dynamic Knowledge Distillation (DKD) method and a noise-elimination strategy for RGB-D Salient Object Detection (SOD). This method adopts an early-fusion strategy to concatenate RGB images and depth images as input, uses a simple VGG16-or VGG19-based Feature Pyramid Network (FPN) as the student model, and dynamically allocates distillation weights by considering the performance of the teacher and the student during the training stage. At the same time, it utilizes the prior knowledge of the teacher network to set a threshold to eliminate the noise of the depth map. Zhang et al. [28] proposed a Multi-layer Semantic Feature Adaptive Distillation (MSFAD) method for object detection. This method uses a routing network and a decision-making agent network composed of a teacher detector and a student detector. Based on the training stage and samples, the student detector can automatically select valuable semantic features from the teacher detector for learning.

Gao et al. [29] proposed a cross-domain few-shot adaptive classification algorithm named SDM based on knowledge distillation technology. The SDM method adopts a teacher-student

architecture and the MixUp technology. It pre-trains in the source domain to enhance the model's learning ability. The model is optimized through weak and strong augmentation operations, as well as supervised classification and alignment distillation training. In the target domain, fine-tuning is carried out through image mixing and a random MixUp strategy to balance the learning difficulty. Zhang et al. [30] proposed an Extreme R-CNN model for few-shot object detection. This model adopts a two-stage fine-tuning approach (TFA), integrating sample synthesis and knowledge distillation techniques. Regarding knowledge distillation, a pre-trained backbone and Feature Pyramid Network (FPN) are utilized as the teacher model to guide the fine-tuning process.

Park et al. [31] proposed a self-knowledge distillation method based on Pixel-Adaptive Label Smoothing (PALS). They generate soft labels by calculating a similarity matrix to aggregate the pixel probability distribution. The final labels are obtained by combining the one-hot encoding distribution and adaptive weights. Moreover, they define the loss function using the KL divergence. Zhang et al. [32] proposed a structured knowledge distillation scheme to address the issues of foreground-background pixel imbalance and the lack of knowledge distillation regarding relationships between pixels, respectively. Chawla et al. [33] proposed a data-free knowledge distillation method for object detection, named DIODE. This method aims to solve the problems that existing knowledge distillation methods in object detection rely on original data or have domain gaps. However, these methods still face some challenges when applied to the detection of missing insulators.

FIAEPI-KD improves edge localization for small targets. It achieves precise multi-scale feature extraction in complex backgrounds. The framework is optimized for real-time insulator missing detection in transmission line scenarios. The FIA module uses dual-path attention. It dynamically enhances multi-scale edge features and suppresses background noise at the feature level. This avoids redundant post-processing. The EPI module introduces gradient-aware distillation loss. It quantifies edge differences between teacher and student models. This ensures feature alignment between restoration and detection tasks.

## Method

### Feature indicator attention

In the realm of transmission line imagery, global context information plays a crucial role. It helps the model quickly locate the region where insulators are absent and distinguish insulators from other background objects. Unlike conventional insulator detection and knowledge distillation methods, the processing pipeline of the proposed approach is shown in Fig 2. Traditional techniques focus on single-level features or use simple network architectures during the feature extraction phase. This study, however, uses a ResNet [34] backbone network with Feature Pyramid Networks (FPN) [35] for multi-scale feature extraction. The feature representation is then refined by the Feature Indicator Attention (FIA) module and Edge Preservation Index (EPI) module. In the knowledge distillation process, traditional methods transfer features directly or use fixed semantic features. This paper, in contrast, introduces a unique attention-guiding and edge-information-retaining mechanism. This mechanism allows for more efficient knowledge transfer from the teacher model to the student model, improving the accuracy and robustness of insulator-missing detection.

The feature maps from the residual blocks in the shallow layers of ResNet mainly contain local edge and texture features of the image. These features are crucial for capturing fine-grained edge information of insulators. In the initial convolutional layers, small 3x3 convolution kernels with a stride of 1 are used to capture the local edge and texture features. As

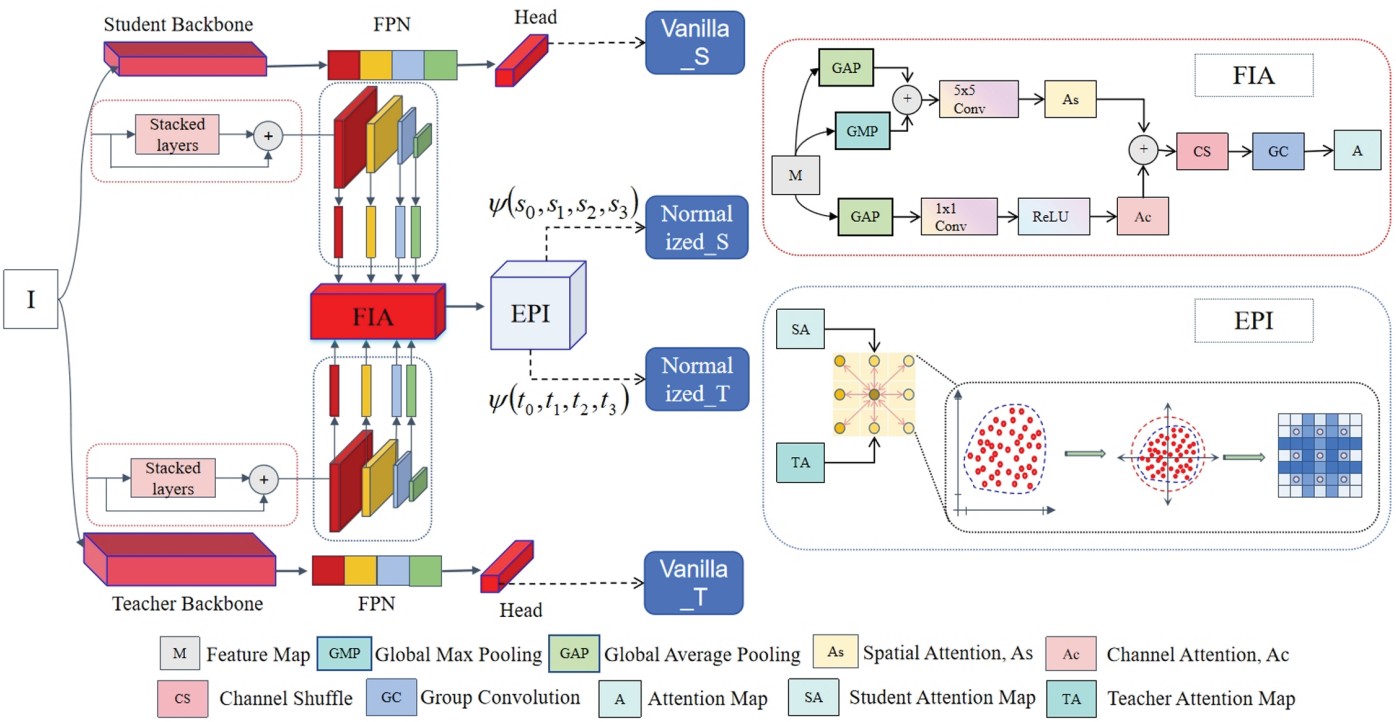

**Fig 2. The proposed knowledge distillation architecture for insulator missing detection: It contains teacher and student networks.** These networks are processed through the backbone, Feature Pyramid Network (FPN), and the head network. There are stacked layers, and the Feature Indicator Attention (FIA) and Edge-Preserving Index (EPI) modules are utilized to interact and normalize the features, generating $Normalized_S$ and $Normalized_T$.

the network depth increases, the feature maps from subsequent residual blocks begin to integrate more high-level semantic information. This includes the overall shape of insulators and their positional relationships with the surrounding environment. To achieve this, larger convolution kernels are used, or the stride is adjusted, which expands the receptive field and captures high-level semantic features like the overall shape and position of the missing insulators in the image.

In the process of multi-scale feature fusion within the FPN, the top feature map is upsampled to match the resolution of the adjacent lower layer. The two feature maps are then combined by addition. The top-layer feature map, despite being smaller, contains more global semantic information. The feature map of the lower layer is slightly larger and holds more local detailed information. This additive fusion method effectively integrates global information and local details. Each fused feature map becomes better at capturing the shape and position of insulators at different scales. The fused feature map can better discern the contours of insulators against complex backgrounds and their variations when captured from different angles and distances. This provides a more detailed and accurate feature representation for subsequent detection tasks.

In this process, the student model ($F^s$) and the teacher model ($F^t$) are obtained concurrently. The global context information $G_s$ and $G_t$ are respectively derived via global average pooling [36] operations. The computational formula for global average pooling is as follows:

$$G_s = \frac{1}{H \times W} \sum_{i=1}^{H} \sum_{j=1}^{W} F^s_{(i,j)} \tag{1}$$

$$G_t = \frac{1}{H \times W} \sum_{i=1}^{H} \sum_{j=1}^{W} F_{(i,j)}^t \tag{2}$$

Where $H$ and $W$ represent the height and width of the feature map, respectively.

To enhance the representation of global context information, we explore the internal correlations among different channels of insulator features. We use $F_{(i,j)}^{s,c}$ and $F_{(i,j)}^{t,c}$ to represent the feature values of the $c$-th channel at position $(i,j)$ in the feature maps of the student and teacher models, respectively. The correlations among channels are captured by calculating the correlation matrices $R_s$ and $R_t$. The computational formulas are as follows:

$$R_s^{c_1,c_2} = \sum_{i=1}^{H} \sum_{j=1}^{W} F_{(i,j)}^{s,c_1} \times F_{(i,j)}^{s,c_2} \tag{3}$$

$$R_t^{c_1,c_2} = \sum_{i=1}^{H} \sum_{j=1}^{W} F_{(i,j)}^{t,c_1} \times F_{(i,j)}^{t,c_2} \tag{4}$$

Among them, $c_1$ and $c_2$ represent different channel indices.

To make the correlation matrices more stable and easier to handle, we normalize them. This gives us the normalized correlation matrices $\hat{R}_s$ and $\hat{R}_t$. After calculating the correlation matrices, we add a normalization layer. The normalization layer normalizes each channel of the correlation matrices, setting the mean to 0 and the variance to 1. This ensures that the model can use the inter-channel correlation information to highlight the feature representation of insulators. Based on the global context information and the normalized correlation matrices, we generate edge-aware attention feature maps to highlight the edges and key regions of insulators. For the student model, the calculation method is as follows:

$$\mathcal{F}_{norm}^s = \sigma(G_s \times \hat{R}_s) \tag{5}$$

where $\sigma$ is the sigmoid function. Similarly, for the teacher model, the calculation method is as follows:

$$\mathcal{F}_{norm}^t = \sigma(G_t \times \hat{R}_t) \tag{6}$$

We combine global context information with inter-channel correlation. The generated attention map reflects the importance of different regions in the image for insulator edge detection.

To focus more precisely on the edge details of insulators and improve model performance, we design a new feature-guided attention module called Feature Indicator Attention (FIA). We use $M \in R^{D \times H \times W}$ to represent the feature map obtained from the above process. Whether it is the feature map of the student model or the teacher model, it is uniformly represented by $M$ when entering this module. This ensures clear logic and universality in the calculation process. Next, we explain the calculation process of the FIA module and its role in improving model performance.

First, calculate the channel attention $A_c$:

$$A_c = \mathcal{F}_{1 \times 1}\left(\max\left(0, \mathcal{F}_{1 \times 1}\left(M_{GAP}^c\right)\right)\right) \tag{7}$$

Here, first perform global average pooling $M_{GAP}^c$ on $M$ in the spatial dimension, then process it through a $1 \times 1$ convolutional layer $\mathcal{F}_{1 \times 1}$, and introduce nonlinearity through the

ReLU [37] activation function. Then, calculate the spatial attention $A_s$:

$$A_s = \mathcal{F}_{5\times5}\left(\left[M_{GMP}^s, M_{GAP}^s\right]\right) \tag{8}$$

This process first performs global max pooling $M_{GMP}^s$ and global average pooling $M_{GAP}^s$ on $M$ in the channel dimension, then concatenates them and processes them through a $5 \times 5$ convolutional layer $\mathcal{F}_{5\times5}$. After that, adjust the channel dimension, first reduce the channel dimension from $D$ to $\frac{D}{p}$ ($p = \frac{D}{8}$) and then restore it to $D$, and fuse them to obtain the coarse attention map $A_{coa} \in \mathbb{R}^{D\times H\times W}$:

$$A_{coa} = A_c + A_s \tag{9}$$

Then refine it through a channel shuffle operation $CS(\cdot)$ and a group convolutional layer $\mathcal{GF}_{5\times5}$ (with $D$ groups), and finally normalize it through a sigmoid operation $\sigma$ to obtain the final attention map $A$:

$$A = \sigma\left(\mathcal{GF}_{5\times5}\left(CS\left(\left[M, A_{coa}\right]\right)\right)\right) \tag{10}$$

### Edge preservation index

In the model, the EPI helps the model learn more accurate edge representations. It does this by quantifying the differences in insulator edge features between the student model and the teacher model. This improves detection accuracy.

The attention map $A$ generated by FAI is applied to the student and teacher feature maps. We take the student feature map $F_s$ as an example. By multiplying the attention map $A$ with the student feature map $F_s$ element-by-element, we obtain the weighted student feature map $F_s^w$.

$$F_s^w = F_s \odot A \tag{11}$$

Then recalculate the weighted edge information. For the calculation of edge information in weighted student feature maps, the calculation method is as follows:

$$s_l^w = \sum_{i=1}^{C}\sum_{j=1}^{N}\left|\mathcal{F}_{norm_{(i,j)}}^{s,w} - \mathcal{F}_{norm_{(i-1,j-1)}}^{s,w}\right| + \left|\mathcal{F}_{norm_{(i,j)}}^{s,w} - \mathcal{F}_{norm_{(i-1,j)}}^{s,w}\right| + \left|\mathcal{F}_{norm_{(i,j)}}^{s,w} - \mathcal{F}_{norm_{(i-1,j+1)}}^{s,w}\right| \tag{12}$$

$$s_m^w = \sum_{i=1}^{C}\sum_{j=1}^{N}\left|\mathcal{F}_{norm_{(i,j)}}^{s,w} - \mathcal{F}_{norm_{(i,j-1)}}^{s,w}\right| + \left|\mathcal{F}_{norm_{(i,j)}}^{s,w} - \mathcal{F}_{norm_{(i,j+1)}}^{s,w}\right| \tag{13}$$

$$s_r^w = \sum_{i=1}^{C}\sum_{j=1}^{N}\left|\mathcal{F}_{norm_{(i,j)}}^{s,w} - \mathcal{F}_{norm_{(i+1,j-1)}}^{s,w}\right| + \left|\mathcal{F}_{norm_{(i,j)}}^{s,w} - \mathcal{F}_{norm_{(i+1,j)}}^{s,w}\right| + \left|\mathcal{F}_{norm_{(i,j)}}^{s,w} - \mathcal{F}_{norm_{(i+1,j+1)}}^{s,w}\right| \tag{14}$$

Here, $s_l^w$ represents the calculation of values in the lower left, left, and upper left directions, $s_m^w$ represents the calculation of values in the directly below and directly above directions, and $s_r^w$ represents the calculation of values in the lower right, right, and upper right directions. Thus, the edge information calculation of the weighted student feature part is expressed as:

$$s^w = s_l^w + s_m^w + s_r^w \tag{15}$$

By calculating $\mathcal{F}_{norm_{(i,j)}}^t$ in the same way, we can obtain $t_l^w$, $t_m^w$, and $t_r^w$, and thus the edge information calculation representation of the teacher feature part is expressed as:

$$t^w = t_l^w + t_m^w + t_r^w \tag{16}$$

The EPI for preserving edge information between student characteristics and teacher characteristics is obtained by the following formula:

$$\mathcal{S}_{\text{epi}}^w = \frac{s^w}{t^w} \tag{17}$$

The EPI for preserving edge information between teacher characteristics and student characteristics is obtained by the following formula:

$$\mathcal{T}_{\text{epi}}^w = \frac{t^w}{s^w} \tag{18}$$

In this way, combining attention guidance with edge-information preservation enhances the model's ability to learn and recognize insulator features. This helps the model more accurately detect the status of insulators in transmission line images.

## Objective function

This paper quantifies the similarity in the feature space by calculating the Euclidean distance between the features $\mathcal{S}_{\text{epi}}$ of the student model and the features $\mathcal{T}_{\text{epi}}$ of the teacher model. In regression tasks, a distance is used to measure the similarity between samples. Unlike traditional distance metrics, the Euclidean distance focuses more on the distance structure of the output.

First, we take a pair of features $\mathcal{S}_{\text{epi}}^w$ from the student model and $\mathcal{T}_{\text{epi}}^w$ from the teacher model. We quantify their similarity by calculating the Euclidean distance between them. We use the distance function $\psi_D$ to measure the Euclidean distance between these two examples in the output representation space.

$$\psi_D(\mathcal{T}_{epi}^i, \mathcal{T}_{epi}^j) = \frac{1}{\omega} \|\mathcal{T}_{epi}^i - \mathcal{T}_{epi}^j\|_2 \tag{19}$$

Here, $\omega$ serves as a normalization factor for the distance. To better focus on the relative distances between other pairs, we set $\omega$ as the average distance between pairs from $\mathcal{X}^2$ in the mini-batch:

$$\omega = \frac{1}{|\mathcal{X}^2|} \sum_{(x_i, x_j) \in \mathcal{X}^2} \|\mathcal{T}_{epi}^i - \mathcal{T}_{epi}^j\|_2 \tag{20}$$

During the knowledge distillation process, there may be scale differences between the teacher distance $\|\mathcal{T}_{epi}^i - \mathcal{T}_{epi}^j\|_2$ and the student distance $\|\mathcal{S}_{epi}^i - \mathcal{S}_{epi}^j\|_2$. This is why the mini-batch distance normalization method is particularly crucial. By leveraging the distance potential measured in both the teacher and the student, we define a distance-potential distillation loss $\mathcal{L}_{dist}$:

$$\mathcal{L}_{dist} = \sum_{(x_i, x_j) \in \mathcal{X}^2} \nu\left(\psi_D(\mathcal{T}_{epi}^i, \mathcal{T}_{epi}^j), \psi_D(\mathcal{S}_{epi}^i, \mathcal{S}_{epi}^j)\right) \tag{21}$$

where $\nu$ is defined as $\begin{cases} 0.5(x_i - y_i)^2 & \text{if } |x_i - y_i| < 1 \\ |x_i - y_i| - 0.5 & \text{otherwise} \end{cases}$.

The distance distillation loss transfers the relationships between instances by penalizing the distance differences between the output representation spaces of instances. Different from traditional KD, it focuses on encouraging the student to pay attention to the distance structure of the output rather than directly matching the teacher's output.

At the same time, we draw on the KL-divergence [38] widely used in previous work for distillation. First, we use the softmax function [39] to convert activations into probability distributions, and then minimize the asymmetric KL-divergence of the normalized activation maps:

$$\mathcal{L}_{KL} = T^2 \sum_{i=1}^{m} \phi(\mathcal{T}_{epi}^i) \cdot \log\left[\frac{\phi(\mathcal{T}_{epi}^i)}{\phi(\mathcal{S}_{epi}^i)}\right] \tag{22}$$

Here, $T$ is a hyperparameter used to control the softness of the target, and $\Phi$ is the softmax function:

$$\phi(\mathcal{T}) = \frac{\exp(\mathcal{T}_{epi}^i/T)}{\sum_{j=1}^{m} \exp(\mathcal{T}_{epi}^j/T)} \tag{23}$$

The loss function $\mathcal{L}_{feat}$ based on the FAI and EPI modules is obtained by the following formula:

$$\mathcal{L}_{feat} = \gamma L_{FAI} + (1 - \gamma) L_{EPI^w} \tag{24}$$

Finally, we use the weighting factors $\alpha$, $\beta$, $\lambda$ to combine $\mathcal{L}_{KL}$, $\mathcal{L}_{dist}$ and $\mathcal{L}_{feat}$, thus obtaining the following overall training objective:

$$\mathcal{L} = \alpha \mathcal{L}_{KL} + \beta \mathcal{L}_{dist} + \lambda \mathcal{L}_{feat} \tag{25}$$

## Experiments

### Implementation details

The UAV-based image acquisition was conducted in publicly accessible areas along transmission lines, where no restricted or protected zones were involved. No permits were required for this study because the fieldwork involved non-invasive observations in a public area, and no endangered or protected species were sampled. All data collection complied with local regulations and ethical guidelines for non-destructive research.

To verify the effectiveness and robustness of our method, we conduct experiments using multiple detection frameworks on the collected insulator-missing dataset. We also use the MSCOCO dataset [40] to verify the generalization of the proposed method. We follow the standard training settings from previous research [41]. The mean average precision $mAP$ is used as the evaluation metric. We report the AP scores at different thresholds and scales: $AP_{50}$, $AP_{75}$, $AP_S$, $AP_M$, and $AP_L$. We compare the widely-used one-stage detector RetinaNet, two-stage detector Faster-RCNN, and anchor-free detector RepPoints. The teacher model uses ResNet/ResNeXt-101, while the student models use ResNet-50 and ResNet18. The corresponding $\lambda$ values are set to 4 and 2, respectively. Each model is trained using the Stochastic Gradient Descent (SGD) optimization algorithm. The momentum is set to 0.9, the weight decay to 1e-4, and the batch size to 2. Our implementation is based on mmdetection and mmrazor in the PyTorch framework. The learning rate of RetinaNet is set to 0.01, while Faster-RCNN and RepPoints have a learning rate of 0.02. The learning rate is reduced by a factor of 10 at the 16th and 22nd epochs, and the training lasts for 24 epochs.

### Main results in custom datasets

In the task of detecting missing insulators in transmission lines, the image samples in the dataset cover scenes shot under different weather conditions, such as sunny, cloudy, rainy, and foggy days. They also include different geographical environments, such as mountains,

plains, and urban peripheries, as well as various angles and distances. These samples contain many instances of normal insulators and insulators with defects such as missing parts, damage, and aging. After pre-processing the image samples, they are input into different models for training and evaluation based on the experimental settings.

We first train teacher models with deep network depths on different detectors. The results are shown in Table 1.

The knowledge distillation strategy based on attention guidance and edge detection uses an attention mechanism that effectively focuses on the key feature regions of insulators in complex backgrounds. For example, in images with obstructions like trees and buildings, it accurately identifies the contour and edge information of insulators. The edge-detection modules enhance the ability to capture subtle changes in the edges of insulators. This allows the model to detect minor defects on the surface of insulators caused by natural erosion or human damage. The results of distillation training for different detector models on the custom dataset are shown in Table 2.

As shown in Fig 3, we present the performance comparison before and after knowledge extraction on a custom dataset in the form of a bar chart. We applied the trained models to the detection task, and the visualization results are shown in Fig 4.

## Main results in COCO datasets

To verify the generalization of the method we proposed, we trained our method on the COCO dataset and compared its performance with several other knowledge distillation techniques. The results are shown in Table 3.

As shown in Fig 5, we present a bar chart comparing the two-stage detectors on the COCO dataset.

**Table 1. Results obtained from pre-training different detector models on the custom dataset.**

| Model | Backbone | $mAP(\%)$ | $AP_{50}(\%)$ | $AP_{75}(\%)$ | $AP_m(\%)$ | $AR_{100}(\%)$ | $AR_{300}(\%)$ | $AR_{1000}(\%)$ |
|---|---|---|---|---|---|---|---|---|
| **Faster-RCNN** [42] | Resnet18 | 42.7 | 87.5 | 37.5 | 42.7 | 51.7 | 51.7 | 51.7 |
| | Resnet50 | 46.9 | 89.4 | 44.2 | 46.9 | 55.4 | 55.4 | 55.4 |
| | Resnet101 | 48.7 | 89.5 | 43.5 | 48.7 | 56.4 | 56.4 | 56.4 |
| **RetinaNet** [43] | Resnet18 | 42.7 | 88.1 | 31.2 | 48.7 | 42.7 | 42.7 | 42.7 |
| | Resnet50 | 54.2 | 92.4 | 58.4 | 54.4 | 61.2 | 61.2 | 61.2 |
| | Resnet101 | 54.7 | 92.4 | 62.4 | 54.7 | 62.2 | 62.2 | 62.2 |
| **Reppoints** [44] | Resnet18 | 49.6 | 91.9 | 52.1 | 49.6 | 56.2 | 56.2 | 56.2 |
| | Resnet50 | 51.9 | 91.4 | 47.2 | 51.9 | 59 | 59 | 59 |
| | Resnet101 | 53.8 | 91.2 | 62 | 53.8 | 59.2 | 59.2 | 59.2 |
| | ResneXt101 | 54.2 | 90.9 | 58.1 | 54.2 | 60.9 | 60.9 | 60.9 |

**Table 2. Results obtained from distillation training of different detector models on the custom dataset.**

| Model | Backbone | $mAP(\%)$ | $AP_{50}(\%)$ | $AP_{75}(\%)$ | $AP_m(\%)$ | Parameters | FLOPS |
|---|---|---|---|---|---|---|---|
| **Faster-RCNN** [42] | Resnet18 | **50.1** | 90.5 | 47.3 | 50.1 | **28.12M** | **140.98** |
| | Resnet50 | **50.4** | 91.0 | 48.7 | 50.4 | **41.53M** | **194.18** |
| | Resnet101 | 48.7 | 89.5 | 43.5 | 48.7 | 60.52M | 270.25 |
| **RetinaNet** [43] | Resnet18 | **53.2** | 93.2 | 60.5 | 53.2 | **21.25M** | **188.74** |
| | Resnet50 | **54.9** | 93.2 | 62.1 | 54.9 | **37.74M** | **239.32** |
| | Resnet101 | 54.7 | 92.4 | 62.4 | 54.7 | 56.74M | 315.39 |
| **Reppoints** [44] | Resnet18 | **57.3** | 91.8 | 70.1 | 57.3 | **20.11M** | **139.24** |
| | Resnet50 | **57.5** | 91.8 | 73.2 | 57.5 | **36.62M** | **190.25** |
| | ResneXt10 1 | 54.2 | 90.9 | 58.1 | 54.2 | 55.62M | 266.32 |

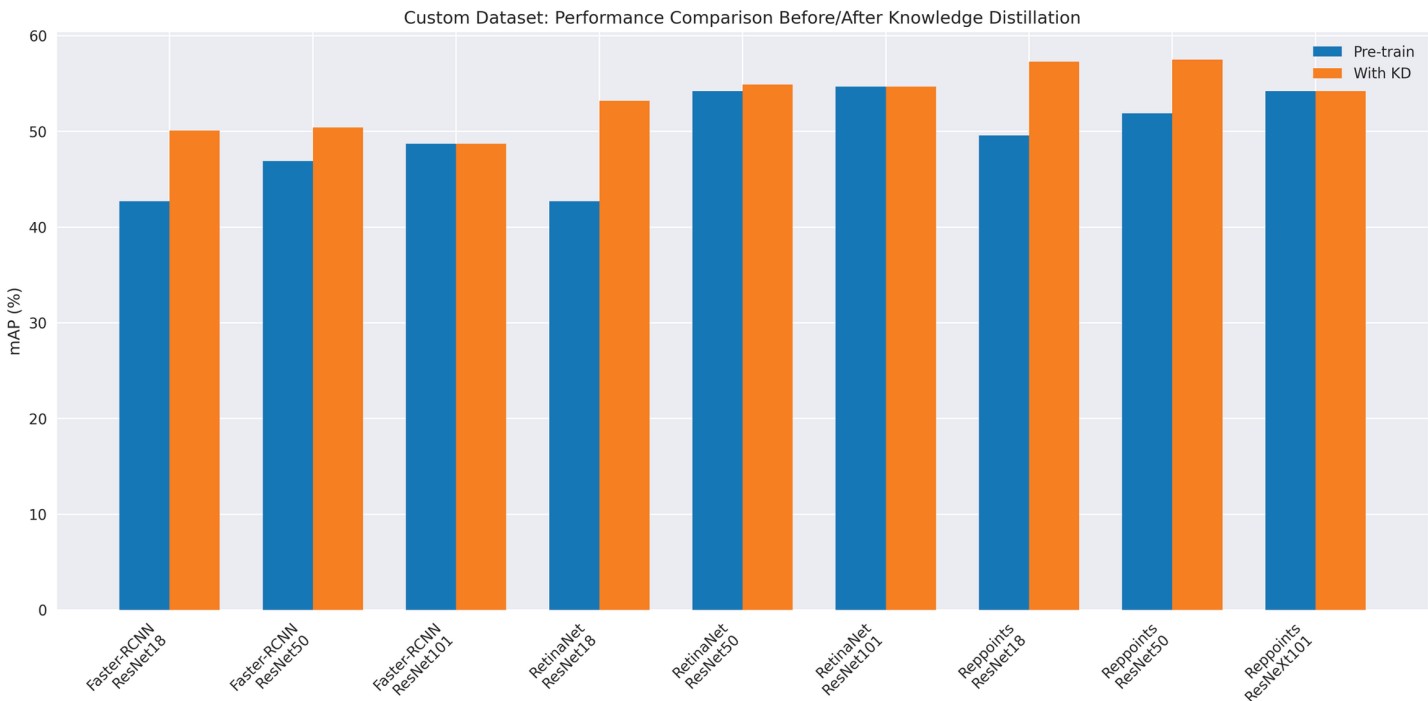

**Fig 3. Custom dataset comparison.**

As shown in Fig 6, we present a multi metric comparison with the state-of-the-art method (two-stage detector) in the form of a radar chart.

On two-stage detectors, the mean average precision ($mAP$) of our method reached 39.8%. This is comparable to the best-performing baseline method. At the $AP_{50}$ threshold, the AP value reached 60.3%. At the $AP_{75}$ threshold, it was 43.2%. The AP scores at different scales also increased. For small objects ($AP_S$), it was 22.1%. For medium objects ($AP_M$), it was 44.0%. For large objects ($AP_L$), it was 52.3%. Compared with the state-of-the-art method FKD, which has an $mAP$ of 39.6%, our method slightly outperforms in overall $mAP$. It also has a significant advantage at $AP_{50}$ (60.3% vs 60.0%). Although FKD has a higher score in $AP_L$ (52.7% vs 52.3%), our method shows competitive performance across all object sizes, demonstrating a balanced improvement.

On one-stage detectors, our method shows competitive performance with an $mAP$ of 40.4%, as shown in Table 4. At the $AP_{50}$ threshold, the AP value reaches 60.1%. At the $AP_{75}$ threshold, it is 43.1%. For different object scales, the $AP_S$ for small objects is 24.1%, the $AP_M$ for medium objects is 45.0%, and the $AP_L$ for large objects is 54.4%. Compared with the best-performing method PKD, which has an $mAP$ of 40.8%, our method shows competitiveness, especially in detecting small objects ($AP_S$: 24.1% vs 23.0%). Although PKD slightly outperforms in terms of overall $mAP$, our method performs outstandingly in specific areas, demonstrating its robustness in different scenarios.

On anchor-free detectors, the mAP of our method is 39.2%. In comparison, the baseline student model has an mAP of 38.6%, while the teacher model has an mAP of 44.2%. We infer that for the key-point-based detection method, since key points are selected before feature extraction, there are fewer computable EPIs. This characteristic enables our method to improve the distillation efficiency, yet the overall performance is still slightly inferior to some other advanced methods.

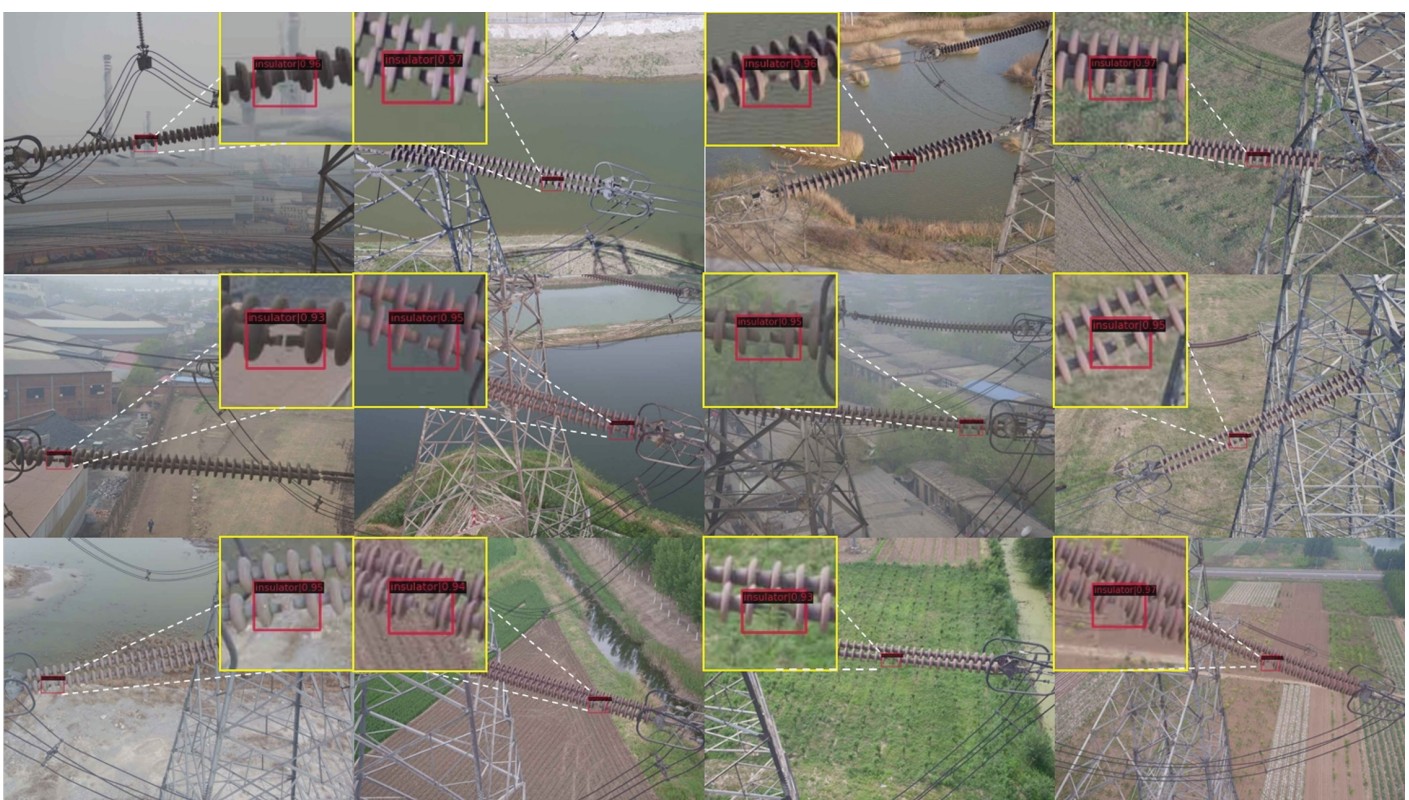

**Fig 4. These are the detection results of the insulator detection model on the dataset.** The model can relatively accurately identify and locate the missing positions of insulators in complex backgrounds such as farmlands, water areas, and factories.

Table 3. The result of two-stage detectors.

| Model | Backbone | $AP(\%)$ | $AP_{50}(\%)$ | $AP_{75}(\%)$ | $AP_S(\%)$ | $AP_M(\%)$ | $AP_L(\%)$ |
|---|---|---|---|---|---|---|---|
| FasterRCNN-Res101 (T) | R50 | 39.8 | 60.1 | 43.3 | 22.5 | 43.6 | 52.8 |
| FasterRCNN-Res50 (S) | R50 | 38.4 | 59.0 | 42.0 | 21.5 | 42.1 | 52.3 |
| Chen [45] | R50 | 38.7 | 59.0 | 42.1 | 22.0 | 41.9 | 51.0 |
| Heo [46] | R50 | 38.9 | 60.1 | 42.6 | 21.8 | 42.7 | 50.7 |
| Fitnet [41] | R50 | 38.9 | 59.5 | 42.4 | 21.9 | 42.2 | 51.6 |
| Wang [47] | R50 | 39.1 | 59.8 | 42.8 | 22.2 | 42.9 | 51.1 |
| FRS [48] | R50 | 39.5 | 60.1 | 43.3 | 22.3 | 43.6 | 51.7 |
| Mimicking [49] | R50 | 39.6 | 60.1 | 43.3 | 22.5 | 42.8 | 52.2 |
| FKD [50] | R50 | 39.6 | 60.0 | 43.0 | 22.9 | 42.9 | 52.7 |
| Song [51] | R50 | 39.6 | 60.0 | 43.1 | — | — | — |
| LD [52] | R50 | 39.8 | 59.1 | 42.1 | — | — | — |
| **Ours** | R50 | **39.8** | **60.3** | 43.2 | 22.1 | **44.0** | 52.3 |

## Ablation study

In this section, we conduct an ablation study to comprehensively evaluate the effectiveness of each module incorporated into our approach. This study involves an analysis of two key aspects, and the results are presented in Table 5.

We further analyze the impact of each module on detecting different types of insulator defects. When only the FIA module is present, the detection accuracy is relatively low.

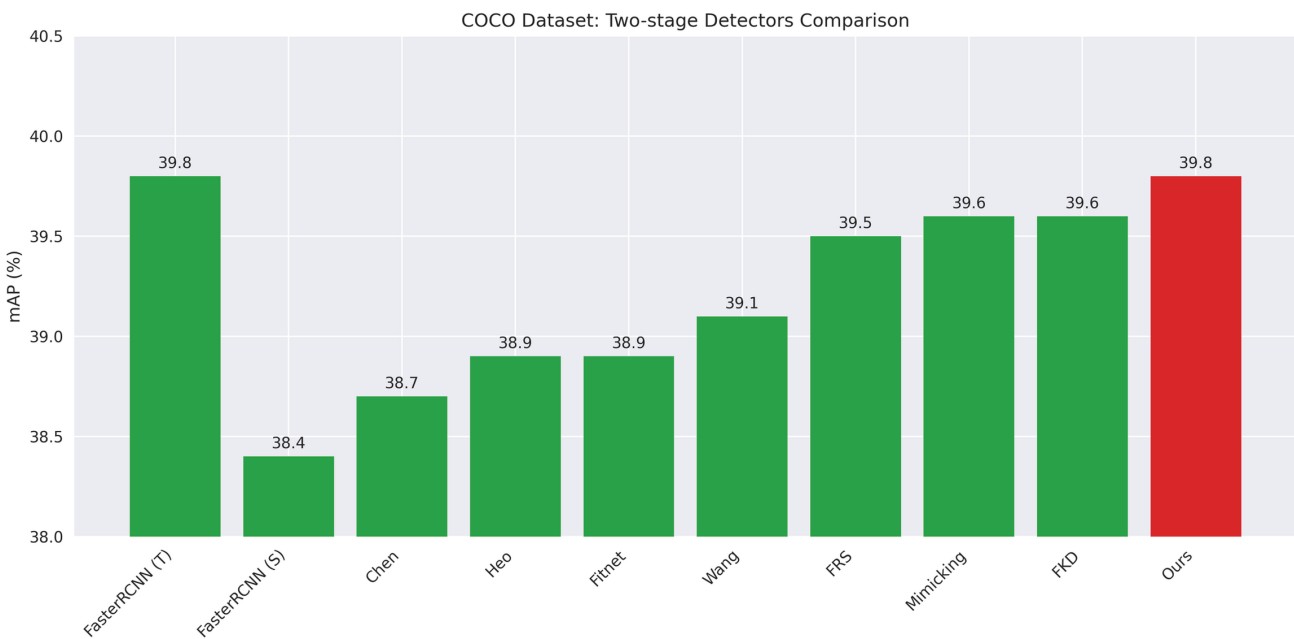

**Fig 5. Comparing the two-stage detectors on the COCO dataset.**

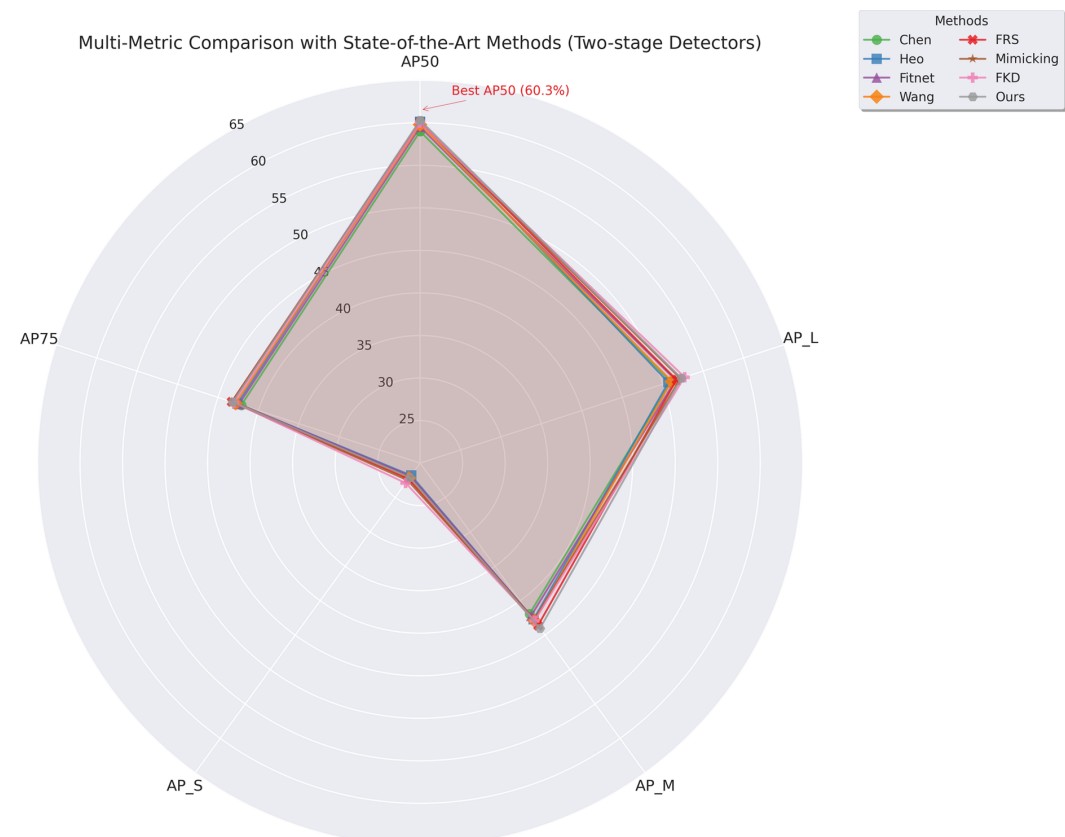

**Fig 6. Multi metric comparison with the state-of-the-art method (two-stage detector).**

**Table 4. The result of one-stage detectors.**

| Model | Backbone | $AP(\%)$ | $AP_{50}(\%)$ | $AP_{75}(\%)$ | $AP_S(\%)$ | $AP_M(\%)$ | $AP_L(\%)$ |
|---|---|---|---|---|---|---|---|
| Retina-ResX101 (T) | R50 | 40.8 | 60.5 | 43.7 | 22.9 | 44.5 | 54.6 |
| Retina-Res50 (S) | R50 | 37.4 | 56.7 | 39.6 | 20.0 | 40.7 | 49.7 |
| Heo [46] | R50 | 37.8 | 58.3 | 41.1 | 21.6 | 41.2 | 48.3 |
| FKD [50] | R50 | 39.6 | 58.8 | 42.1 | 22.7 | 43.3 | 52.5 |
| SKD [32] | R50 | 39.6 | 58.8 | 42.1 | 22.7 | 43.3 | 52.5 |
| FRS [48] | R50 | 40.1 | 59.5 | 42.5 | 21.9 | 43.7 | 54.3 |
| FGD [53] | R50 | 40.4 | 59.9 | 43.3 | 23.4 | 44.7 | 54.1 |
| SSIM [54] | R50 | 40.6 | 59.7 | 43.7 | 23.6 | 44.8 | 53.9 |
| Shu [55] | R50 | 40.8 | 60.4 | 43.4 | 22.7 | 44.5 | 55.3 |
| PKD [56] | R50 | 40.8 | 60.3 | 43.4 | 23.0 | 45.1 | 54.7 |
| **Ours** | R50 | **40.4** | 60.1 | 43.1 | **24.1** | 45.0 | 54.4 |

**Table 5. The results of ablation studies.**

| Model | Backbone | $AP(\%)$ | $AP_{50}(\%)$ | $AP_{75}(\%)$ | $AP_S(\%)$ | $AP_M(\%)$ | $AP_L(\%)$ |
|---|---|---|---|---|---|---|---|
| Retina-ResX101 (T) | R50 | 40.8 | 60.5 | 43.7 | 22.9 | 44.5 | 54.6 |
| Retina-Res50 (S) | R50 | 37.4 | 56.7 | 39.6 | 20.0 | 40.7 | 49.7 |
| +*FIA* | R50 | 38.3 | 53.7 | 40.7 | 20.6 | 41.6 | 51.7 |
| +*FIA+EPI* | R50 | **40.4** | **60.1** | **43.1** | **24.1** | **45.0** | **54.4** |

This may be because it can focus on key areas but lacks the ability to capture edge details accurately. After adding the EPI module, the detection accuracy improves significantly. This indicates that the Edge-Preservation Information module is crucial for detecting minor defects. For detecting large-area damage defects, the FIA module can quickly locate the damaged area. However, when used alone, it may be affected by background interference. After adding the EPI module, it can better identify the damaged edges and improve detection accuracy. Through these analyses, we gain a deeper understanding of each module's role in detecting different defect types.

As shown in Fig 7, we present the effectiveness of the ablation experimental research component in the form of a bar chart.

## Conclusion

This article proposes FIEPI-KD, a novel knowledge distillation method that integrates FIA and EPI for detecting missing insulators in transmission lines. FIEPI-KD uses a dual-path spatial channel attention mechanism to focus on key insulator regions and quantify the edge differences between teacher and student models. It has achieved significant improvements in various detector architectures. The framework outperforms state-of-the-art distillation methods on our custom dataset and MSCOCO benchmark testing. The ablation study validated the roles of FIA and EPI.

The combination of attention-guided feature alignment and edge-aware distillation has created a new paradigm for knowledge transfer in object detection. In theory, FIEPI-KD bridges the performance gap between lightweight student models and computationally intensive teacher models. This provides new ideas for complex background and multi-scale object detection. In practice, the safety monitoring of transmission lines has been improved, reducing the risk of power outages and economic losses from insulator failures.

Although progress has been made, this work lacks coverage for extreme weather conditions, such as heavy snow, and ultra-small insulator defects.

### Ablation Study: Component Effectiveness

**Fig 7. The effectiveness of the ablation experimental research component.**

Future research can enhance the model's adaptability to extreme environments. The dataset can be expanded to include rare defects, occlusion, and adversarial conditions, such as extreme lighting and snow. Another direction is to expand the use of FIEPI-KD for defect detection in other power equipment.

## Acknowledgments

We would like to acknowledge the support of the Scientific Research Innovation Team Construction Project of Qingdao City University (Grant No. QCU23TDKJ01).

## Author contributions

**Conceptualization:** Hanzhi Cui, Dawei Huang.

**Data curation:** Dawei Huang.

**Formal analysis:** Qiuxue Ouyang.

**Investigation:** Hanzhi Cui.

**Methodology:** Dawei Huang.

**Resources:** Hanzhi Cui, Conghan Zhong.

**Software:** Hanzhi Cui, Wancheng Feng.

**Supervision:** Conghan Zhong.

**Validation:** Wancheng Feng, Zhengao Li.

**Visualization:** Zhengao Li.

**Writing – original draft:** Hanzhi Cui, Dawei Huang.

**Writing – review & editing:** Dawei Huang, Wancheng Feng, Conghan Zhong.

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
