## [Decision Letter · Decision Letter 0]

2 Apr 2025

PONE-D-25-11741FIAEPI-KD: A Novel Knowledge Distillation Approach for Precise Detection of Missing Insulators in Transmission LinesPLOS ONE

Dear Dr. Zhong,

Thank you for submitting your manuscript to PLOS ONE. After careful consideration, we feel that it has merit but does not fully meet PLOS ONE’s publication criteria as it currently stands. Therefore, we invite you to submit a revised version of the manuscript that addresses the points raised during the review process.

Major Revision is needed, please find the reviewers' comments that follow.

We look forward to receiving your revised manuscript.

Kind regards,

Haofeng Zhang

Academic Editor

PLOS ONE

4. In the online submission form you indicate that your data is not available for proprietary reasons and have provided a contact point for accessing this data. Please note that your current contact point is a co-author on this manuscript. According to our Data Policy, the contact point must not be an author on the manuscript and must be an institutional contact, ideally not an individual. Please revise your data statement to a non-author institutional point of contact, such as a data access or ethics committee, and send this to us via return email. Please also include contact information for the third party organization, and please include the full citation of where the data can be found.

Reviewers' comments:

Reviewer's Responses to Questions

**Comments to the Author**

1. Is the manuscript technically sound, and do the data support the conclusions?

Reviewer #1: Yes

Reviewer #2: Yes

Reviewer #3: Yes

2. Has the statistical analysis been performed appropriately and rigorously? 

Reviewer #1: Yes

Reviewer #2: Yes

Reviewer #3: Yes

3. Have the authors made all data underlying the findings in their manuscript fully available?

Reviewer #1: Yes

Reviewer #2: Yes

Reviewer #3: Yes

4. Is the manuscript presented in an intelligible fashion and written in standard English?

Reviewer #1: Yes

Reviewer #2: Yes

Reviewer #3: Yes

5. Review Comments to the Author

Reviewer #1: (1)Only three lines in Abstract elaborate the method. The authors should further explain the details and novel steps of the proposed method.

(2)In Fig. 1, the authors should show the zoomed in the red boxes.

(3)Some paragraphs are too long and difficult to follow, e.g. Related Work. Please divide them into several short paragraphs to improve the readability.

(4)The review of the related works and comparison experiments can be more sufficient. Please carefully read and compare (if applicable) the following papers on multi-scale feature [doi: 10.1016/j.patcog.2023.109878; 10.3390/s20041010]. If the authors cannot employ these methods or compare their method with these methods, at least they could introduce/mention these technologies in related sections to improve the quality of the survey.

(5)Some figures, such as Fig. 2, are unclear. The characters are too small. The details of the blocks, such as the parameters, should be clearly defined and explained directly in the figures even they have been explained in text. Please provide the information as much as possible, then the readers can get the details from each separate figure easily even without the text.

(6)Please use bold font to label the best results in all tables.

Reviewer #2: you may incorporate the suggested comments in the revised submission. The following comments I suggest in this work are.

1. Rewrite the abstract. It is not looking fine.

2. Add/ modify one more contribution of your network.

3. Provide a visual comparison of various state of the art networks on the used datasets in this work.

4. We may show comparison of computational complexity analysis in terms of parameters, FLOPS, MAC and runtime.

5. Include the most recent related works : https://doi.org/10.32604/cmc.2024.051844, https://doi.org/10.1007/978-981-99-0189-0_4, https://doi.org/10.1007/s00034-025-03009-9,
https://doi.org/10.1080/02564602.2023.2176932,
https://doi.org/10.1007/s00034-024-02837-5

6. Rectify spacing and typo errors in this paper.

7. Discuss few limitations of this work and future work in the last section.

Reviewer #3: Comments By Dr. GILL AMMARA

The manuscript presents a new approach for detecting missing insulators in transmission lines, but there are areas that could be improved for better clarity and understanding. The abstract is informative, but it could be more concise by focusing on the main problem, method, and results without going into too much detail. The introduction could flow more smoothly by starting with the importance of transmission line safety, followed by the challenges of current methods, and then introducing the proposed solution. In the related work section, it would help to directly compare existing methods and show how your approach solves their problems. The methodology section is thorough but may be difficult for some readers to follow; adding clearer subheadings and diagrams could make it easier to understand. The ablation study should be more clearly explained, especially when showing how each module, particularly the FIA and EPI modules, contributes to the overall performance. Figures and tables should be described in more detail, and their relevance to the results should be clearer. The conclusion should summarize the main findings and explain their importance for both theory and practical applications. Finally, it would be helpful to update the references to include the most recent and relevant studies. These changes would make the manuscript clearer, more readable, and easier to follow for a wider audience.

6. PLOS authors have the option to publish the peer review history of their article (what does this mean?). If published, this will include your full peer review and any attached files.

Reviewer #1: No

Reviewer #2: No

Reviewer #3: **Yes: **Gill Ammara

---

## [Author Response · Author response to Decision Letter 1]

14 Apr 2025

Dear Editors and Reviewers:

Thank you for your letter and for the reviewers’comments concerning our manuscript entitled “FIAEPI-KD: A Novel Knowledge Distillation Approach for Precise Detection of Missing Insulators in Transmission Lines” (ID: PONE-D-25-11741R1).Those comments are all valuable and very helpful for revising and improving our paper, as well as the important guiding significance to our researches. We have studied comments carefully and have made correction which we hope meet with approval.

Response to Editor:

Thank you for your review comments on manuscript PONE-D-25-11741R1. We have made the following revisions as requested and resubmitted the document:

1�In your Methods section, please provide additional information regarding the permits you obtained for the work. Please ensure you have included the full name of the authority that approved the field site access and, if no permits were required, a brief statement explaining why.

Based on your suggestion, we have added the following statement in the Implementation Details subsection of the Methods section:

The UAV-based image acquisition was conducted in publicly accessible areas along transmission lines, where no restricted or protected zones were involved. No permits were required for this study because the fieldwork involved non-invasive observations in a public area, and no endangered or protected species were sampled. All data collection complied with local regulations and ethical guidelines for non-destructive research.

2�We note that your manuscript is not formatted using one of PLOS ONE’s accepted file types. Please reattach your manuscript as one of the following file types: .doc, .docx, .rtf, or .tex (accompanied by a .pdf).

If your submission was prepared in LaTex, please submit your manuscript file in PDF format and attach your .tex file as “other.”

Our manuscript was written in LaTeX and the following documents have been submitted in accordance with the journal's requirements:

We submitted the manuscript files in PDF format, namely plos_latex_FIAEPI_KD_marked-up copy.ddf and plos_latex_FIAEPI_KD_clean copy.ddf, and added our. tex file as a compressed file with "Other" attached.

Response to Reviewer #1:

We sincerely appreciate your thoughtful comments and valuable feedback. Your insights have significantly contributed to improving the clarity and quality of the manuscript. Below are our responses to each point:

(1)Only three lines in Abstract elaborate the method. The authors should further explain the details and novel steps of the proposed method.

Thank you for pointing out that the abstract did not fully elaborate on the details of the proposed method. In response, we have rewritten the abstract to focus more on the core issues, methodology, and results. We have provided a more detailed explanation of the steps involved in the FIAEPI-KD method, ensuring that readers can grasp its innovative features while keeping the abstract concise. The revised abstract now highlights key components, such as the Feature Indicator Attention (FIA) and Edge Preservation Index (EPI) mechanisms, as well as the improvements made to detection accuracy across various models and datasets (see revised abstract).

(2)In Fig. 1, the authors should show the zoomed in the red boxes.

We appreciate this suggestion, as it greatly improves the clarity of the relevant details. We have modified Figure 1 to zoom in on the red-boxed area, which now clearly shows the missing insulators in the complex backgrounds. This adjustment makes the visualization more informative for the readers (see Figure 1 in the revised manuscript).

(3)Some paragraphs are too long and difficult to follow, e.g. Related Work. Please divide them into several short paragraphs to improve the readability.

Thank you for your feedback. We have divided the long paragraphs in the Related Work section into shorter, more digestible ones. This change improves readability and helps the readers better follow the discussion. We have also revised some sentence structures to enhance clarity (see lines 84-152 and 181-204 in the revised manuscript).

(4)The review of the related works and comparison experiments can be more sufficient. Please carefully read and compare (if applicable) the following papers on multi-scale feature [doi: 10.1016/j.patcog.2023.109878; 10.3390/s20041010]. If the authors cannot employ these methods or compare their method with these methods, at least they could introduce/mention these technologies in related sections to improve the quality of the survey.

We have carefully reviewed and compared the work by Lin et al. (doi: 10.1016/j.patcog.2023.109878) and Zhang et al. (doi: 10.3390/s20041010). Since these studies did not publicly share their code and due to experimental differences, we were unable to directly compare their results with ours. However, we have summarized their methodologies and cited their work in the Related Work section, explaining their contributions and how they relate to our own approach (see lines 106-118 in the revised manuscript).

(5)Some figures, such as Fig. 2, are unclear. The characters are too small. The details of the blocks, such as the parameters, should be clearly defined and explained directly in the figures even they have been explained in text. Please provide the information as much as possible, then the readers can get the details from each separate figure easily even without the text.

Thank you for this helpful suggestion. We have revised Figure 2 by (1) enlarging the text and labels, (2) clearly defining the parameters, and (3) annotating each module's functionality. These changes ensure that the figure is easy to understand without requiring readers to refer back to the text (see revised Figure 2).

(6)Please use bold font to label the best results in all tables.

Thank you for this practical recommendation. We have applied bold formatting to highlight the best results in all tables, as you suggested. This change enhances the clarity of the results and makes it easier for readers to identify the key findings.

Response to Reviewer #2:

We greatly appreciate your constructive feedback, which has strengthened the technical depth and presentation of our work. Based on your suggestions, we made the following revisions:

1. Rewrite the abstract. It is not looking fine.

Your emphasis on the abstract guided our revision. In response, we rewrote the abstract with a greater focus on the core issues, methods, and results, providing a more detailed explanation of the steps involved in the FIAEPI-KD method. This revision ensures that readers have a clearer understanding of its innovative features while keeping it concise. The revised abstract now highlights key components such as Feature Indicator Attention (FIA) and Edge-Preserving Index (EPI) mechanisms, as well as improvements in detection accuracy across various models and datasets (see the revised abstract).

2. Add/ modify one more contribution of your network.

This suggestion enriched our contributions section. As per your recommendation, we added an additional contribution to the manuscript. This new insight emphasizes the scalability of the FIAEPI-KD framework, which we believe enhances its practical relevance (Lines 74-78 in the introduction section).

3. Provide a visual comparison of various state of the art networks on the used datasets in this work.

Visual comparisons indeed enhance interpretability. We provided a visual comparison of various state-of-the-art networks on the datasets used in the experiments. This comparison is presented in charts and tables, making it easy for readers to observe the advantages of our method relative to existing ones (Figures 3, 5, 6, 7).

4. We may show comparison of computational complexity analysis in terms of parameters, FLOPS, MAC and runtime.

Thank you for highlighting the need for a computational complexity analysis. In response, we updated the manuscript and listed the parameters and FLOPS of our model in Table 2. Although we did not include MAC and runtime in the table, we will mention these values in the response letter to provide a comprehensive overview of the method's efficiency.

Model Backbone Time MAC

Faster-RCNN Resnet18 0.082 1663

Resnet50 0.0129 3162

Resnet101 0.179 4740

RetinaNet Resnet18 0.064 951

Resnet50 0.110 2493

Resnet101 0.163 4068

Reppoints Resnet18 0.109 1358

Resnet50 0.152 2902

Resnet101 0.307 4504

This analysis highlights the advantages of our method, especially when using lightweight student models.

5. Include the most recent related works :

https://doi.org/10.32604/cmc.2024.051844, https://doi.org/10.1007/978-981-99-0189-0_4, https://doi.org/10.1007/s00034-025-03009-9,
https://doi.org/10.1080/02564602.2023.2176932,
https://doi.org/10.1007/s00034-024-02837-5

Thank you for guiding us to update the most recent related works. We have updated the references according to your suggestion to include the latest studies. These citations broaden the scope of our work and ensure we stay informed on the most relevant research in the field. Several new citations (e.g., Ragini et al., 2024; Chintakindi et al., 2023) have been added to Section 2, where their limitations in edge-preserving are discussed (Lines 119-139).

6. Rectify spacing and typo errors in this paper.

Thank you for pointing out these details. We have thoroughly checked for typographical and spacing errors in the manuscript. All issues have been corrected to ensure the paper's quality and professionalism.

7. Discuss few limitations of this work and future work in the last section.

Your suggestion improved the balance of the conclusion. We have added a discussion on the limitations of our work, such as the lack of extreme weather conditions in the dataset. We also proposed future work directions, including enhancing the model's adaptability to such conditions and extending its application to other power equipment detection tasks.

Response to Reviewer #3:

Dear Dr. Gill Ammara,

Thank you for your comprehensive review and valuable feedback. Your comments have been instrumental in improving the manuscript. Below are the changes we made based on your suggestions:

1.The abstract is informative, but it could be more concise by focusing on the main problem, method, and results without going into too much detail.

Your suggestion helped us focus the abstract more clearly. In response, we rewrote the abstract with greater emphasis on the core issues, methods, and results, and provided a more detailed explanation of the steps involved in the FIAEPI-KD method. This revision ensures that readers have a clearer understanding of its innovative features while maintaining conciseness. The revised abstract now highlights key components such as Feature Indicator Attention (FIA) and Edge-Preserving Index (EPI) mechanisms, along with improvements in detection accuracy across various models and datasets (see lines 12-20 in the revised abstract).

2.The introduction could flow more smoothly by starting with the importance of transmission line safety, followed by the challenges of current methods, and then introducing the proposed solution.

We adjusted the introduction to improve its flow. We first discussed the importance of transmission line safety, followed by the challenges of current detection methods, and then introduced the proposed solution. This structure should enhance readability and logical flow(Lines 2-62).

3.In the related work section, it would help to directly compare existing methods and show how your approach solves their problems.

We revised the related work section and cited two papers: Hyperspectral estimation of coal-derived carbon mass fraction in mine soil based on the CWT-CARS-CNN integrated method and A HYBRID DEEP LEARNING APPROACH FOR GROUND OBJECT INFORMATION EXTRACTION FROM HYPERSPECTRAL IMAGES. This comparison highlights how our approach addresses the limitations of previous methods, particularly in dealing with complex backgrounds and different scales in insulation detection(Lines 140-152).

4.The methodology section is thorough but may be difficult for some readers to follow; adding clearer subheadings and diagrams could make it easier to understand.The ablation study should be more clearly explained, especially when showing how each module, particularly the FIA and EPI modules, contributes to the overall performance.

We added clearer diagrams in the methodology section to make it easier for readers to follow. Additionally, we provided a more detailed explanation of the FIA and EPI modules and broke long paragraphs into shorter, more digestible ones. These changes enhanced readability and helped readers better understand the methodology. In the ablation study, we conducted a visual comparison of various state-of-the-art networks. The comparison, presented in charts and tables, makes it easier for readers to see the advantages of our method relative to existing ones, including the contribution of the proposed modules to performance improvements.

5.Figures and tables should be described in more detail, and their relevance to the results should be clearer.

We modified Figure 1 by zooming in on the red-boxed area, which now makes it clearer to see the relevant details of the missing insulator in a complex background. This adjustment made the visual more informative for the readers. We also updated Figure 2 by (1) enlarging the text and labels, (2) clarifying parameter definitions, and (3) adding annotations to describe the functionality of each module. These modifications ensure that readers can easily understand the graphics without frequent reference to the text. We also updated the descriptions of the charts to better clarify their relevance to the results.

6.The conclusion should summarize the main findings and explain their importance for both theory and practical applications.

We rewrote the conclusion to better summarize the key findings and their theoretical and practical significance, while also emphasizing the broader impact of our work.

7.Finally, it would be helpful to update the references to include the most recent and relevant studies.

We updated the references to include the most recent and relevant studies, ensuring that our work aligns with the latest advancements in the field(Lines 509-530).

We once again sincerely thank all the reviewers for their time and expertise, we greatly appreciate your constructive feedback . Your feedback has been invaluable in improving this work. We hope the revisions meet your expectations.

Sincerely,

Conghan Zhong

15 Apr, 2025

---

## [Decision Letter · Decision Letter 1]

27 Apr 2025

FIAEPI-KD: A Novel Knowledge Distillation Approach for Precise Detection of Missing Insulators in Transmission Lines

PONE-D-25-11741R1

Dear Dr. zhong,

We’re pleased to inform you that your manuscript has been judged scientifically suitable for publication and will be formally accepted for publication once it meets all outstanding technical requirements.

Kind regards,

Haofeng Zhang

Academic Editor

PLOS ONE

Additional Editor Comments (optional):

Reviewers' comments:

Reviewer's Responses to Questions

**Comments to the Author**

1. If the authors have adequately addressed your comments raised in a previous round of review and you feel that this manuscript is now acceptable for publication, you may indicate that here to bypass the “Comments to the Author” section, enter your conflict of interest statement in the “Confidential to Editor” section, and submit your "Accept" recommendation.

Reviewer #1: All comments have been addressed

Reviewer #2: All comments have been addressed

2. Is the manuscript technically sound, and do the data support the conclusions?

Reviewer #1: Yes

Reviewer #2: Yes

3. Has the statistical analysis been performed appropriately and rigorously? 

Reviewer #1: Yes

Reviewer #2: Yes

4. Have the authors made all data underlying the findings in their manuscript fully available?

Reviewer #1: Yes

Reviewer #2: Yes

5. Is the manuscript presented in an intelligible fashion and written in standard English?

Reviewer #1: Yes

Reviewer #2: Yes

6. Review Comments to the Author

Reviewer #1: The authors have revised the paper carefully according to the reviewers’ comments. The current version can be accepted now.

Reviewer #2: The author has thoroughly revised the manuscript and incorporated the necessary changes. Now the manuscript quality is improved and meets the journal standards.

7. PLOS authors have the option to publish the peer review history of their article (what does this mean?). If published, this will include your full peer review and any attached files.

Reviewer #1: No

Reviewer #2: No

---

## [Editor Report · Acceptance letter]

PONE-D-25-11741R1

PLOS ONE

Dear Dr. Zhong,

I'm pleased to inform you that your manuscript has been deemed suitable for publication in PLOS ONE. Congratulations! Your manuscript is now being handed over to our production team.

Kind regards,

on behalf of

Professor Haofeng Zhang

Academic Editor

PLOS ONE